# COMPRESSIVE TRANSFORMERS FOR LONG-RANGE SEQUENCE MODELLING

**Jack W. Rae**⁂†‡   **Anna Potapenko\*†**   **Siddhant M. Jayakumar†**   **Chloe Hillier†**

**Timothy P. Lillicrap**†‡

## ABSTRACT

We present the Compressive Transformer, an attentive sequence model which compresses past memories for long-range sequence learning. We find the Compressive Transformer obtains state-of-the-art language modelling results in the WikiText-103 and Enwik8 benchmarks, achieving 17.1 ppl and 0.97 bpc respectively. We also find it can model high-frequency speech effectively and can be used as a memory mechanism for RL, demonstrated on an object matching task. To promote the domain of long-range sequence learning, we propose a new open-vocabulary language modelling benchmark derived from books, PG-19.

## 1 INTRODUCTION

Humans have a remarkable ability to remember information over long time horizons. When reading a book, we build up a compressed representation of the past narrative, such as the characters and events that have built up the story so far. We can do this even if they are separated by thousands of words from the current text, or long stretches of time between readings. During daily life, we make use of memories at varying time-scales: from locating the car keys, placed in the morning, to recalling the name of an old friend from decades ago. These feats of memorisation are not achieved by storing every sensory glimpse throughout one's lifetime, but via lossy compression. We aggressively select, filter, or integrate input stimuli based on factors of surprise, perceived danger, or repetition — amongst other signals (Richards and Frankland, 2017).

Memory systems in artificial neural networks began with very compact representations of the past. Recurrent neural networks (RNNs, Rumelhart et al. (1986)) learn to represent the history of observations in a compressed state vector. The state is *compressed* because it uses far less space than the history of observations — the model only preserving information that is pertinent to the optimization of the loss. The LSTM (Hochreiter and Schmidhuber, 1997) is perhaps the most ubiquitous RNN variant; it uses learned gates on its state vector to determine what information is stored or forgotten from memory.

However since the LSTM, there has been great benefit discovered in *not* bottlenecking all historical information in the state, but instead in keeping past activations around in an external memory and *attending* to them. The Transformer (Vaswani et al., 2017) is a sequence model which stores the hidden activation of every time-step, and integrates this information using an attention operator (Bahdanau et al., 2014). The Transformer will thus represent the past with a tensor (depth × memory size × dimension) of past observations that is, in practice, an order of magnitude larger than an LSTM's hidden state. With this granular memory, the Transformer has brought about a step-change in state-of-the-art performance, within machine translation (Vaswani et al., 2017), language modelling (Dai et al., 2019; Shoeybi et al., 2019), video captioning (Zhou et al., 2018), and a multitude of language understanding benchmarks (Devlin et al., 2018; Yang et al., 2019) amongst others.

One drawback in storing everything is the computational cost of attending to every time-step and the storage cost of preserving this large memory. Several works have focused on reducing the computational cost of attention with sparse access mechanisms (Rae et al., 2016; Child et al., 2019;

---

⁂Authors contributed equally, † DeepMind, London, UK. ‡ CoMPLEX, Computer Science, University College London, UK. Please direct correspondence to {jwrae, apotapenko}@google.com.

Sukhbaatar et al., 2019; Lample et al., 2019). However sparse attention does not solve the storage problem, and often requires custom sparse kernels for efficient implementation. Instead we look back to the notion of compactly representing the past. We show this can be built with simple dense linear-algebra components, such as convolutions, and can reduce both the space and compute cost of our models.

We propose the Compressive Transformer, a simple extension to the Transformer which maps past hidden activations (memories) to a smaller set of compressed representations (compressed memories). The Compressive Transformer uses the same attention mechanism over its set of memories and compressed memories, learning to query both its short-term granular memory and longer-term coarse memory. We observe this improves the modelling of text, achieving state-of-the-art results in character-based language modelling — 0.97 bpc on Enwik8 from the Hutter Prize (Hutter, 2012) — and word-level language modelling — 17.1 perplexity on WikiText-103 (Merity et al., 2016). Specifically, we see the Compressive Transformer improves the modelling of rare words.

We show the Compressive Transformer works not only for language, but can also model the waveform of high-frequency speech with a trend of lower likelihood than the TransformerXL and Wavenet (Oord et al., 2016) when trained over 400,000 steps. We also show the Compressive Transformer can be used as a memory component within an RL agent, IMPALA (Espeholt et al., 2018), and can successfully compress and make use of past observations.

Furthermore we present a new book-level language-modelling benchmark PG-19, extracted from texts in Project Gutenberg[1], to further promote the direction of long-context sequence modelling. This is over double the size of existing LM benchmarks and contains text with much longer contexts.

## 2 RELATED WORK

There have been a variety of recent attempts to extend the range of attention, particularly in the Transformer, or to replace the attention operation with something less expensive. Wu et al. (2019) show that a convolution-like operator that runs in linear time can actually exceed the performance of the quadratic-time self-attention layer in the Transformer at sentence-to-sentence translation and sentence-level language modelling. However such a mechanism inhibits the flow of information across a large number of time-steps for a given layer, and has not shown to be beneficial for long-range sequence modelling.

Dai et al. (2019) propose the TransformerXL, which keeps past activations around in memory. They also propose a novel relative positional embedding scheme which they see outperforms the Transformer's original absolute positional system. Our model incorporates both of these ideas, the use of a memory to preserve prior activations and their relative positional embedding scheme.

The Sparse Transformer (Child et al., 2019) uses fixed sparse attention masks to attend to roughly $\sqrt{n}$ locations in memory. This approach still requires keeping all memories around during training, however with careful re-materialization of activations and custom kernels, the authors are able to train the model with a reasonable budget of memory and compute. When run on Enwik8, the much larger attention window of $8,000$ improves model performance, but overall it does not significantly outperform a simpler TransformerXL with a much smaller attention window.

The use of dynamic attention spans is explored in Sukhbaatar et al. (2019). Different attention heads can learn to have shorter or longer spans of attention — and they observe this achieves state-of-the-art in character-based language modelling. This idea could easily be combined with our contribution — a compressive memory. However an efficient implementation is not possible on current dense-linear-algebra accelerators, such as Google's TPUs, due to the need for dynamic and sparse computation. Our approach builds on simple dense linear algebra components, such as convolutions.

## 3 MODEL

We present the Compressive Transformer, a long-range sequence model which compacts past activations into a compressed memory[2]. The Compressive Transformer is a variant of the Transformer

---

[1]Project Gutenberg: `https://www.gutenberg.org/`
[2]A TF implementation can be found in Sonnet: `https://github.com/deepmind/sonnet`

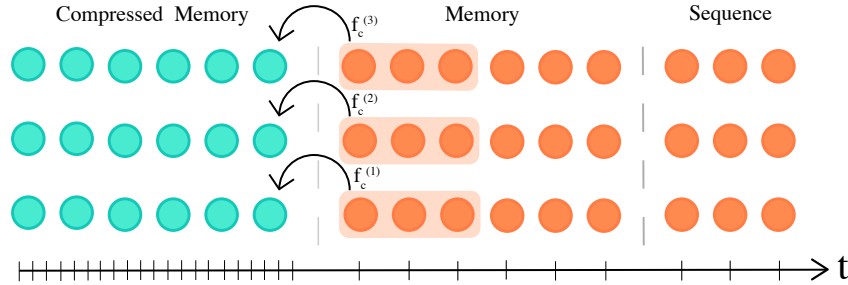

Figure 1: The Compressive Transformer keeps a fine-grained memory of past activations, which are then compressed into coarser *compressed* memories. The above model has three layers, a sequence length $n_s = 3$, memory size $n_m = 6$, compressed memory size $n_{cm} = 6$. The highlighted memories are compacted, with a compression function $f_c$ per layer, to a single compressed memory — instead of being discarded at the next sequence. In this example, the rate of compression $c = 3$.

(Vaswani et al., 2017), a deep residual network which only uses attention to propagate information over time (namely *multi-head attention*). We build on the ideas of the TransformerXL (Dai et al., 2019) which maintains a memory of past activations at each layer to preserve a longer history of context. The TransformerXL discards past activations when they become sufficiently old (controlled by the size of the memory). The key principle of the Compressive Transformer is to compress these old memories, instead of discarding them, and store them in an additional *compressed memory*.

## 3.1 DESCRIPTION

We define $n_m$ and $n_{cm}$ to be the number of respective memory and compressive memory slots in the model per layer. The overall input sequence $\mathcal{S} = x_1, x_2, \ldots, x_{|s|}$ represents input observations (e.g. tokens from a book). These are split into fixed-size windows of size $n_s$ for the model to process in parallel. The model observes $\mathbf{x} = x_t, \ldots, x_{t+n_s}$ at time $t$, which we refer to as the *sequence* (e.g. in Figure 1). As the model moves to the next sequence, its $n_s$ hidden activations are pushed into a fixed-sized FIFO memory (like the TransformerXL) of size $n_m$. The oldest $n_s$ activations in memory are evicted, but unlike the TransformerXL we do not discard them. Instead we apply a *compression operation*, $f_c : \mathbf{R}^{n_s \times d} \to \mathbf{R}^{\lfloor \frac{n_s}{c} \rfloor \times d}$, mapping the $n_s$ oldest memories to $\lfloor \frac{n_s}{c} \rfloor$ compressed memories which we then store in a secondary FIFO *compressed memory* of size $n_{cm}$. $d$ denotes the hidden size of activations and $c$ refers to the compression rate, a higher value indicates more coarse-grained compressed memories. The overall temporal range of the model becomes $l \times (n_s + n_m + c * n_{cm})$, where $l$ is the number of layers — as discussed in Supplementary Section A. The full architecture is described in Algorithm 1.

---

**Algorithm 1** Compressive Transformer

---

At time zero
1: $\mathbf{m_0} \leftarrow \mathbf{0}$       // Initialize memory to zeros ($l \times n_m \times d$)
2: $\mathbf{cm_0} \leftarrow \mathbf{0}$       // Initialize compressed memory to zeros ($l \times n_{cm} \times d$)
At time t
3: $\mathbf{h}^{(1)} \leftarrow \mathbf{x W_{emb}}$       // Embed input sequence($n_s \times d$)
4: **for** layer $i = 1, 2, \ldots, l$ **do**
5:     $\mathbf{mem^{(i)}} \leftarrow \text{concat}(\mathbf{cm_t^{(i)}}, \mathbf{m_t^{(i)}})$       // (($n_{cm} + n_m) \times d$)
6:     $\mathbf{\tilde{a}^{(i)}} \leftarrow \text{multihead\_attention}^{(i)}(\mathbf{h^{(i)}}, \mathbf{mem_t^{(i)}})$       // MHA over both mem types ($n_s \times d$)
7:     $\mathbf{a^{(i)}} \leftarrow \text{layer\_norm}(\mathbf{\tilde{a}^{(i)}} + \mathbf{h^{(i)}})$       // Regular skip + layernorm ($n_{cm} \times d$)
8:     $\mathbf{old\_mem^{(i)}} \leftarrow \mathbf{m_t^{(i)}}[: n_s]$       // Oldest memories to be forgotten ($n_s \times d$)
9:     $\mathbf{new\_cm^{(i)}} \leftarrow f_c^{(i)}(\mathbf{old\_mem^{(i)}})$       // Compress oldest memories by factor $c$ ($\lfloor \frac{n_s}{c} \rfloor \times d$)
10:     $\mathbf{m_{t+1}^{(i)}} \leftarrow \text{concat}(\mathbf{m_t^{(i)}}, \mathbf{h^{(i)}})[-n_m :]$       // Update memory ($n_m \times d$)
11:     $\mathbf{cm_t^{(i)}} \leftarrow \text{concat}(\mathbf{cm_t^{(i)}}, \mathbf{new\_cm^{(i)}})[-n_{cm} :]$       // Update compressed memory ($n_{cm} \times d$)
12:     $\mathbf{h^{(i+1)}} \leftarrow \text{layer\_norm}(\text{mlp}^{(i)}(\mathbf{a^{(i)}}) + \mathbf{a^{(i)}})$       // Mixing MLP ($n_s \times d$)

---

**Algorithm 2** Attention-Reconstruction Loss

1: $L^{attn} \leftarrow 0$
2: **for** layer $i = 1, 2, \ldots, l$ **do**
3:    $\mathbf{h^{(i)}} \leftarrow$ stop_gradient($\mathbf{h^{(i)}}$)               // Stop compression grads from passing...
4:    $\mathbf{old\_mem^{(i)}} \leftarrow$ stop_gradient($\mathbf{old\_mem^{(i)}}$)        // ...into transformer network.
5:    $\mathbf{Q}, \mathbf{K}, \mathbf{V} \leftarrow$ stop_gradient(attention params at layer i)     // Re-use attention weight matrices.
6:    def attn($\mathbf{h}, \mathbf{m}$) $\leftarrow \sigma((\mathbf{hQ})(\mathbf{mK}))(\mathbf{mV})$        // Use content-based attention (no relative).
7:    $\mathbf{new\_cm^{(i)}} \leftarrow f_c^{(i)}(\mathbf{old\_mem^{(i)}})$          // Compression network (to be optimized).
8:    $L^{attn} \leftarrow L^{attn} + ||\text{attn}(\mathbf{h^{(i)}}, \mathbf{old\_mem^{(i)}}) - \text{attn}(\mathbf{h^{(i)}}, \mathbf{new\_cm^{(i)}})||_2$

## 3.2 COMPRESSION FUNCTIONS AND LOSSES

For choices of compression functions $f_c$ we consider **(1) max/mean pooling**, where the kernel and stride is set to the compression rate $c$;     **(2) 1D convolution** also with kernel & stride set to $c$; **(3) dilated convolutions**; **(4) most-used** where the memories are sorted by their average attention (usage) and the most-used are preserved. The pooling is used as a fast and simple baseline. The *most-used* compression scheme is inspired from the garbage collection mechanism in the Differentiable Neural Computer (Graves et al., 2016) where low-usage memories are erased. The convolutional compression functions contain parameters which require training.

One can train the compression network using gradients from the loss; however for very old memories this requires backpropagating-through-time (**BPTT**) over long unrolls. As such we also consider some local auxiliary compression losses. We consider an **auto-encoding** loss where we reconstruct the original memories from the compressed memories $\mathcal{L}^{ae} = ||\mathbf{old\_mem^{(i)}} - g(\mathbf{new\_cm^{(i)}})||_2$, where $g : \mathbb{R}^{\frac{n_s}{c} \times d} \rightarrow \mathbb{R}^{n_s \times d}$ is learned. This is a lossless compression objective — it attempts to retain all information in memory. We also consider an **attention-reconstruction** loss described in Algorithm 2 which reconstructs the content-based attention over memory, with content-based attention over the compressed memories. This is a lossy objective, as information that is no longer attended to can be discarded, and we found this worked best. We stop compression loss gradients from passing into the main network as this prevents learning. Instead the Transformer optimizes the task objective and the compression network optimizes the compression objective conditioned on task-relevant representations; there is no need to mix the losses with a tuning constant.

# 4 PG-19 BENCHMARK

As models begin to incorporate longer-range memories, it is important to train and benchmark them on data containing larger contexts. Natural language in the form of text provides us with a vast repository of data containing long-range dependencies, that is easily accessible. We propose a new language modelling benchmark, **PG-19**, using text from books extracted from Project Gutenberg [3]. We select Project Gutenberg books which were published over 100 years old, i.e. before 1919 (hence the name PG-19) to avoid complications with international copyright, and remove short texts. The dataset contains $28,752$ books, or $11GB$ of text — which makes it over double the size of BookCorpus and Billion Word Benchmark.

## 4.1 RELATED DATASETS

The two most benchmarked word-level language modelling datasets either stress the modelling of stand-alone sentences (Billion Word Benchmark from Chelba et al. (2013)) or the modelling of a small selection of short news articles (Penn Treebank processed by Mikolov et al. (2010)). Merity et al. (2016) proposed the WikiText-103 dataset, which contains text from a high quality subset of English-language wikipedia articles. These articles are on average $3,600$ words long. This dataset has been a popular recent LM benchmark due to the potential to exploit longer-range dependencies (Grave et al., 2016; Rae et al., 2018; Bai et al., 2018b). However recent Transformer models, such

---

[3]PG-19 is available at `https://github.com/deepmind/pg19`

Table 1: Comparison to existing popular language modelling benchmarks.

|  | Avg. length (words) | Train Size | Vocab | Type |
|---|---|---|---|---|
| 1B Word | 27 | 4.15GB | 793K | News (sentences) |
| Penn Treebank | 355 | 5.1MB | 10K | News (articles) |
| WikiText-103 | 3.6K | 515MB | 267K | Wikipedia (articles) |
| PG-19 | 69K | 10.9GB | (open) | Books |

as the TransformerXL (Dai et al., 2019) appear to be able to exploit temporal dependencies on the order of several thousand words. This motivates a larger dataset with longer contexts.

Books are a natural choice of long-form text, and provide us with stylistically rich and varied natural language. Texts extracted from books have been used for prior NLP benchmarks; such as the Children's Book Test (Hill et al., 2015) and LAMBADA (Paperno et al., 2016). These benchmarks use text from Project Gutenberg, an online repository of books with expired US copyright, and Book-Corpus (Zhu et al., 2015), a prior dataset of $11K$ unpublished (at time of authorship) books. CBT and LAMBADA contain extracts from books, with a specific task of predicting held-out words. In the case of LAMBADA the held-out word is specifically designed to be predictable for humans with access to the full textual context — but difficult to guess with only a local context.

CBT and LAMBADA are useful for probing the linguistic intelligence of models, but are not ideal for training long-range language models from scratch as they truncate text extracts to at most a couple of paragraphs, and discard a lot of the books' text. There has been prior work on training models on book data using BookCorpus directly (e.g. BERT from Devlin et al. (2018)) however BookCorpus is no longer distributed due to licensing issues, and the source of data is dynamically changing — which makes exact benchmarking difficult over time.

The NarrativeQA Book Comprehension Task (Kočiský et al., 2018) uses Project Gutenberg texts paired with Wikipedia articles, which can be used as summaries. Due to the requirement of needing a corresponding summary, NarrativeQA contains a smaller selection of books: 1,527 versus the 28,752 books in PG-19. However it is reasonable that PG-19 may be useful for pre-training book summarisation models.

## 4.2 STATISTICS

A brief comparison of PG-19 to other LM datasets can be found in Table 1. We intentionally do not limit the vocabulary by *unk-ing* rare words, and release the dataset as an open-vocabulary benchmark. To compare models we propose to continue measuring the word-level perplexity. This can still be computed for any chosen character-based, byte-based or subword-based scheme. To do this, one calculates the total cross-entropy loss $L = -\sum_t \log(p_t|p_{<t})$ over the given validation or test subset using a chosen tokenization scheme, and then one normalizes this value by the number of words: $L/n_{words}$ where $n_{words}$ is the total number of words in the given subset, taken from Table 2. The word-level perplexity is thus $e^{L/n_{words}}$. For sake of model comparisons, it is important to use the exact number of words computed in Table 2 as the normalisation constant.

Alongside quantitative analyses, we build an LDA topic model (Blei et al., 2003) for a qualitative inspection of the text. We present key words for several topics in the Supplementary Table 10. These topics include art, education, naval exploration, geographical description, war, ancient civilisations, and more poetic topics concerning the human condition — love, society, religion, virtue etc. This contrasts to the more objective domains of Wikipedia and news corpora.

## 5 EXPERIMENTS

We optimised all models with Adam (Kingma and Ba, 2014). We used a learning rate schedule with a linear warmup from 1e-6 to 3e-4 and a cosine decay back down to 1e-$n$6. For character-based LM we used $4,000$ warmup steps with $100,000$ decay steps, and for word-based LM we used $16,000$ warmup steps with $500,000$ decay steps. We found that decreasing the optimisation update frequency helped (see Section 5.5.1), namely we only applied parameter updates every 4 steps after $60,000$ iterations. However we found the models would optimise well for a range of warmup/warm-

Table 2: PG-19 statistics split by subsets.

|          | Train         | Valid.     | Test      |
|----------|---------------|------------|-----------|
| # books  | 28,602        | 50         | 100       |
| # words  | 1,973,136,207 | 3,007,061  | 6,966,499 |

Table 3: Eval. perplexities on PG-19.

|                          | Valid. | Test |
|--------------------------|--------|------|
| 36L TransformerXL        | 45.5   | 36.3 |
| **36L Compressive Transf.** | 43.4   | 33.6 |

Table 4: State-of-the-art results on Enwik8.

| Model                                      | BPC   |
|--------------------------------------------|-------|
| 7L LSTM (Graves, 2013)                     | 1.67  |
| LN HyperNetworks Ha et al. (2016)          | 1.34  |
| LN HM-LSTM Chung et al. (2016)             | 1.32  |
| ByteNet (Kalchbrenner et al., 2016)        | 1.31  |
| RHN Zilly et al. (2017)                    | 1.27  |
| mLSTM Krause et al. (2016)                 | 1.24  |
| 64L Transf. Al-Rfou et al. (2019)          | 1.06  |
| 24L TXL (Dai et al., 2019)                 | 0.99  |
| Sparse Transf. (Child et al., 2019)        | 0.991 |
| Adaptive Transf. (Sukhbaatar et al., 2019) | 0.98  |
| *24L TXL (ours)*                           | 0.98  |
| 24L Compressive Transformer                | **0.97** |

Table 5: Compression approaches on Enwik8.

| Compression fn | Compression loss | BPC   |
|----------------|------------------|-------|
| Conv           | BPTT             | 0.996 |
| Max Pooling    | N/A              | 0.986 |
| Conv           | Auto-encoding    | 0.984 |
| Mean Pooling   | N/A              | 0.982 |
| Most-used      | N/A              | 0.980 |
| Dilated conv   | Attention        | 0.977 |
| Conv           | Attention        | **0.973** |

down values. We clipped the gradients to have a norm of at most $0.1$, which was crucial to successful optimisation.

## 5.1 PG-19

We benchmark the Compressive Transformer against the TransformerXL on the newly proposed PG-19 books dataset. Because it is open-vocabulary, we train a subword vocabulary of size 32000 with SubwordTextEncoder from the tfds package in TensorFlow and use the dataset statistics to compute word-level perplexity, as described in Section 4.2. We train a 36 layer Compressive Transformer with a window size of $512$, both memory and compressed memory size of $512$, and compression rate $C = 2$. We compare this to a 36 layer TransformerXL trained with window size $512$ and attention window $1024$. The model was trained on 256 TPUv3 cores with a total batch size of $512$ and converged after processing around 100 billion subword tokens. We display the results in Table 3 where we see the Compressive Transformer obtains a test perplexity of $33.6$ versus the TransformerXL's $36.3$. Despite the dataset size, it is clearly a challenging domain. This can suit as a first baseline on the proposed long-range language modelling benchmark. We show samples from this model in Supplementary Section F. The model is able to generate long-form narrative of varying styles: from character dialogue, first person diary entries, to descriptive third-person text.

## 5.2 ENWIK8

We compare the TransformerXL and the Compressive Transformer on the standard character-level language modelling benchmark Enwik8 taken from the Hutter Prize (Hutter, 2012), which contains 100M bytes of unprocessed Wikipedia text. We select the first 90MB for training, 5MB for validation, and the latter 5MB for testing — as per convention. We train 24-layer models with a sequence window size of 768. During training, we set the TransformerXL's memory size to 2304, and for the Compressive Transformer we use memory of size 768 and compressed memory of size 1152 with compression rate $C = 3$. During evaluation, we increased the TransformerXL memory size to 4096 and the compressed memory in our model to 3072 (after sweeping over the validation set), obtaining the numbers reported in Table 4. We show the effect of scaling the compressed memory size and evaluation performance in Supplementary Section C. The proposed model achieves the new state-of-the-art on this dataset with $0.97$ bits-per-character.

We compare compression functions and the use of auxiliary losses in Table 5. We sweep over compression rates of $2$, $3$, and $4$ and report results with the best performing value for each row. BPTT signifies that no auxiliary compression loss was used to train the network other than the

Table 6: Validation and test perplexities on WikiText-103.

|  | Valid. | Test |
|---|---|---|
| LSTM (Graves et al., 2014) | - | 48.7 |
| Temporal CNN (Bai et al., 2018a) | - | 45.2 |
| GCNN-14 (Dauphin et al., 2016) | - | 37.2 |
| Quasi-RNN Bradbury et al. (2016) | 32 | 33 |
| RMC (Santoro et al., 2018) | 30.8 | 31.9 |
| LSTM+Hebb. (Rae et al., 2018) | 29.0 | 29.2 |
| Transformer (Baevski and Auli, 2019) | - | 18.7 |
| 18L TransformerXL, M=384 (Dai et al., 2019) | - | 18.3 |
| *18L TransformerXL, M=1024 (ours)* | - | 18.1 |
| 18L Compressive Transformer, M=1024 | **16.0** | **17.1** |

overall training loss. To feed gradients into the compression function we unrolled the model over double the sequence length and halved the batch size to fit the larger unroll into memory.

## 5.3 WIKITEXT-103

We train an eighteen-layered Compressive Transformer on the closed-vocabulary word-level language modelling benchmark WikiText-103, which contains articles from Wikipedia. We train the model with a compressed memory size, memory size, and a sequence window size all equal to $512$. We trained the model over $64$ Tensor Processing Units (TPU) v3 with a batch size of 2 per core — making for a total batch size of $128$. The model converged in a little over 12 hours. We found the single-layer convolution worked best, with a compression rate of $c = 4$. This model obtained $17.6$ perplexity on the test set. By tuning the memory size over the validation set — setting the memory size to $500$, and compressed memory size to $1,500$ — we obtain $17.1$ perplexity. This is $1.2$ perplexity points over prior state of the art, and means the model places a $\approx 5\%$ higher probability on the correct word over the prior SotA TransformerXL.

It is worth noting that in Table 6 we do not list methods that use additional training data, or that make use of test-time labels to continue training the model on the test set (known as dynamic evaluation (Graves, 2013)). If we incorporate a very naive dynamic evaluation approach of loading a model checkpoint and continuing training over one epoch of the test set, then we obtain a test perplexity of **16.1**. This is slightly better than the published 16.4 from Krause et al. (2019) — which uses a more sophisticated dynamic evaluation approach on top of the TransformerXL. However in most settings, one does not have access to test-time labels — and thus we do not focus on this setting. Furthermore there has been great progress in showing that more data equates to much better language modelling; Shoeybi et al. (2019) find a large transformer 8B-parameter transformer trained on 170GB of text obtains 10.7 word-level perplexity on WikiText-103. However it is not clear to what extent the WikiText-103 test set may be leaked inside these larger training corpora. For clarity of model comparisons, we compare to published results trained on the WikiText-103 training set.

We break perplexity down by word frequency in Table 7 and see the Compressive Transformer makes only a small modelling improvement for frequent words (2.6% over the TransformerXL baseline) but obtains a much larger improvement of $\approx 20\%$ for infrequent words. Furthermore, we see **10X** improvement in modelling rare words over the prior state-of-the-art LSTM language model published in 2018 — which demonstrates the rate of progress in this area.

## 5.4 COMPRESSIBILITY OF LAYERS

We can use compression to better understand the model's mode of operation. We inspect how compressible Transformer's activations are as they progress through higher layers in the network. One may expect representations to become more difficult to compress at higher layers, if more semantic information is represented there. We monitor the compression loss at each layer of our best-performing Compressive Transformer models trained on Enwik8 and WikiText-103 and display these in Supplementary Section B Figure 6. We note that the compression loss is about one order of magnitude higher for word-level language modelling (WikiText-103) over character-level langauge

Table 7: WikiText-103 test perplexity broken down by word frequency buckets. The most frequent bucket is words which appear in the training set more than $10,000$ times, displayed on the left. For reference, a uniform model would have perplexity $|V| = 2.6e5$ for all frequency buckets. *LSTM comparison from Rae et al. (2018)

|  | $> 10K$ | $1K-10K$ | $100-1K$ | $< 100$ | All |
|---|---|---|---|---|---|
| LSTM* | 12.1 | 219 | 1,197 | 9,725 | 36.4 |
| TransformerXL (ours) | 7.8 | 61.2 | 188 | 1,123 | 18.1 |
| Compressive Transformer | **7.6** | **55.9** | **158** | **937** | **17.1** |
| Relative gain over TXL | 2.6% | 9.5% | 21% | 19.9% | 5.8% |

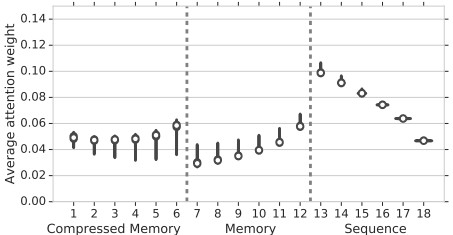
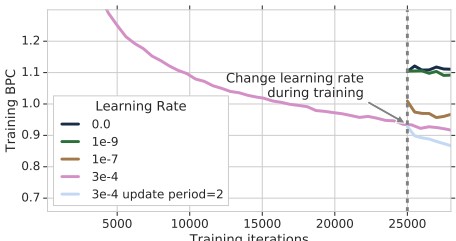

Figure 2: **Attention weight on Enwik8**. Average attention weight from the sequence over the compressed memory (oldest), memory, and sequence (newest) respectively. The sequence self-attention is causally masked, so more attention is placed on earlier elements in the sequence. There is an increase in attention at the transition from memory to compressed memory.

Figure 3: **Learning rate analysis**. Reducing the learning rate (e.g. to zero) during training (on Enwik8) harms training performance. Reducing the frequency of optimisation updates (effectively increasing the batch size) is preferable.

modelling (Enwik8). Furthermore the first layer of the Transformer is highly compressible. However there is not a clear trend of compression cost increasing with layer depth.

## 5.5 ATTENTION

We inspect where the network is attending to on average, to determine whether it is using its compressed memory. We average the attention weight over a sample of $20,000$ sequences from a trained model on Enwik8. We aggregate the attention into eighteen buckets, six for each of the compressed memory, memory, and sequence respectively. We set the size of the sequence, memory and compressed memory all to be $768$. We plot this average attention weight per bucket in Figure 2 with a $1\sigma$ standard error. We see most of the attention is placed on the current sequence; with a greater weight placed on earlier elements of the sequence due to the causal self-attention mechanism which masks future attention weights. We also observe there is an *increase* in attention from the oldest activations stored in the regular memory, to the activations stored in the compressed memory. **This goes against the trend of older memories being accessed less frequently — and gives evidence that the network is learning to preserve salient information**.

### 5.5.1 OPTIMISATION SCHEDULE

We make an observation about an interesting but undesirable meta-learning phenomenon during long-context training. When the learning rate is tuned to be much smaller (or set to zero) during training, performance degrades drastically both for the TransformerXL and the Compressive Transformer. This is displayed in Figure 3.

Usually we consider distributional shift from the training data to the test data, but we can also observe a shift in the model when transferring from a training to evaluation mode (even when the model is evaluated on the training data). In this case, this is due to the online updating of parameters whilst processing long contiguous articles. We would like the model to generalise well to scenarios where it is not continuously optimised. Updating the parameters only at article boundaries (and then

resetting the state) could be one solution for long-range memory models, but this would slow down learning significantly.

Instead, we propose reducing the frequency of optimisation updates during training. We find this allows for the best of both worlds — fast initial learning with frequent updates, and better generalisation near the end of training with less frequent updates (e.g. every 4 steps). Reducing the optimisation frequency increases the effective batch size, which has also been shown to be preferable to learning rate decay in image modelling (Smith et al., 2018). We observed a final performance improvement in our TransformerXL baseline on Enwik8, from $0.995$ — which approximately replicates the published result — to $0.984$ — which matches the most recent SotA architecture. We note, the additional space and compute cost of accumulating gradients is negligible across iterations, so there was no performance regression in using this scheme.

## 5.6 SPEECH

We train the Compressive Transformer on the waveform of speech to assess its performance on different modalities. Speech is interesting because it is sampled at an incredibly high frequency, but we know it contains a lot of information on the level of phonemes and entire phrases.

To encourage long-term reasoning, we refrain from conditioning the model on speaker identity or text features, but focus on unconditional speech modelling. We train the model on 24.6 hours of 24kHz North American speech data. We chunk the sequences into windows of size 3840, roughly 80ms of audio, and compare a 20-layer Compressive Transformer to a 20-layer TransformerXL and a 30-layer WaveNet model (Oord et al., 2016) — a state-of-the-art audio generative model used to serve production speech synthesis applications at Google (Oord et al., 2018). All networks have approximately 40M parameters, as WaveNet is more parameter-efficient per layer. We train each network with 32 V100 GPUs, and a batch size of 1 per core (total batch size of 32) using synchronous training.

WaveNet processes an entire chunk in parallel, however the TransformerXL and Compressive Transformer are trained with a window size of 768 and a total memory size of $1,568$ (for the Compressive Transformer we use 768 memory + 768 compressed). We thus unroll the model over the sequence. Despite this sequential unroll, the attention-based models train at only half the speed of WaveNet. We see the test-set negative-log-likelihood in Figure 4, and observe that a Compressive Transformer with a compression rate of $4$ is able to outperform the TransformerXL and maintain a slim advantage over WaveNet. However we only trained models for at most one week (with 32GPUs) and it would be advantageous to continue training until full convergence — before definitive conclusions are made.

## 5.7 REINFORCEMENT LEARNING

Compression is a good fit for video input sequences because subsequent frames have high mutual information. Here we do not test out the Compressive Transformer on video, but progress straight to a reinforcement learning agent task that receives a video stream of visual observations — but must ultimately learn to use its memory to reason over a policy.

We test the Compressive Transformer as a drop-in replacement to an LSTM in the IMPALA setup (Espeholt et al., 2018). Otherwise, we use the same training framework and agent architecture as described in the original work with a fixed learning rate of 1.5e-5 and entropy cost coefficient of 2e-3. We test the Compressive Transformer on a challenging memory task within the DMLab-30 (Beattie et al., 2016) domain, *rooms_select_nonmatching_object*. This requires the agent to explore a room in a visually rich 3D environment and remember the object present. The agent can then advance to a second room where it must select the object *not present* in the original room. This necessitates that the agent both remember events far in the past, and also learn to efficiently reason about them.

We fix both the memory and compressed memory sizes to $64$. In Figure 5, we present results for a range of compression rates, averaged over 3 seeds. We see that the best performing agents endowed with the Compressive Transformer are able to solve the task to human-level. We note that the model with compression rate $1$ is unable to learn the task to the same proficiency. The speed of learning and stability seem to increase proportionally with higher rates of compression (up to a limit) – i.e.

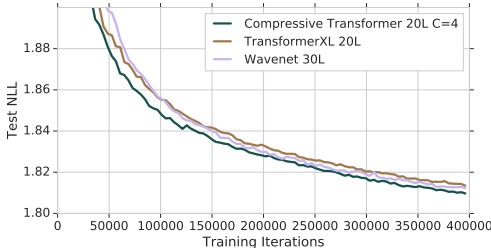

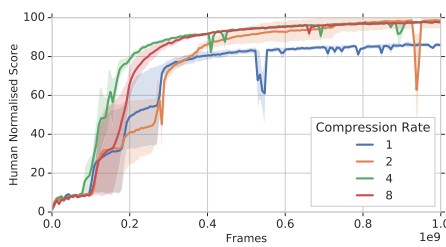

Figure 4: **Speech Modelling.** We see the Compressive Transformer is able to obtain competitive results against the state-of-the-art WaveNet in the modelling of raw speech sampled at 24kHz.

Figure 5: **Vision and RL**. We see the Compressive Transformer integrates visual information across time within an IMPALA RL agent, trained on an object matching task.

the effective memory window of the agent – and we find compression rate $4$ to once again be the best performing. We see this as a promising sign that the architecture is able to efficiently learn, and suitably use, compressed representations of its visual input and hope to test this more widely in future work.

## 6  CONCLUSION

In this paper we explore the notion of compression as a means of extending the temporal receptive field of Transformer-based sequence models. We see a benefit to this approach in the domain of text, with the Compressive Transformer outperforming existing architectures at long-range language modelling. To continue innovation in this area, we also propose a new book-level LM benchmark, PG-19. This may be used to compare long-range language models, or to pre-train on other long-range reasoning language tasks, such as NarrativeQA (Kočiský et al., 2018).

We see the idea of compressive memories is applicable not only to the modality of text, but also audio, in the form of modelling the waveform of speech, and vision, within a reinforcement-learning agent trained on a maze-like memory task. In both cases, we compare to very strong baselines (Wavenet (Oord et al., 2016) and IMPALA (Espeholt et al., 2018)).

The main limitation of this work is additional complexity, if the task one wishes to solve does not contain long-range reasoning then the Compressive Transformer is unlikely to provide additional benefit. However as a means of scaling memory and attention, we do think compression is a simpler approach to dynamic or sparse attention — which often requires custom kernels to make efficient. One can build effective compression modules from simple neural network components, such as convolutions. The compression components are immediately efficient to run on GPUs and TPUs.

Memory systems for neural networks began as compressed state representations within RNNs. The recent wave of progress using attention-based models with deep and granular memories shows us that it is beneficial to refrain from immediately compressing the past. However we hypothesise that more powerful models will contain a mixture of granular recent memories and coarser compressed memories. Future directions could include the investigation of adaptive compression rates by layer, the use of long-range shallow memory layers together with deep short-range memory, and even the use of RNNs as compressors. Compressive memories should not be forgotten about just yet.

## ACKNOWLEDGEMENTS

We thank Chris Dyer, Felix Gimeno, and Koray Kavukcuoglu for reviewing the manuscript. We thank Peter Dayan, Adam Santoro, Jacob Menick, Emilio Parisotto, Hyunjik Kim, Simon Osindero, Sergey Bartunov, David Raposo, and Daan Wierstra for ideas regarding model design. We thank Yazhe Li and Aaron Van de Oord for their help and advice in instrumenting speech modelling experiments. Finally, we thank our wider DeepMind colleagues for supporting this project with stimulating discussions, engineering infrastructure, and positive reinforcement signals.

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

# SUPPLEMENTARY MATERIALS

## A  TEMPORAL RANGE OF THE COMPRESSIVE TRANSFORMER

The TransformerXL with a memory of size $n$ has a maximum temporal range of $l \times n$ with an attention cost of $\mathcal{O}(n_s^2 + n_s n)$ (see Dai et al. (2019) for a detailed discussion). The Compressive Transformer now has a maximum temporal range of $l \times (n_s + n_m + c * n_{cm})$ with an attention cost of $\mathcal{O}(n_s^2 + n_s(n_m + n_{cm}))$. For example, setting $n_{cm} = n_m = n/2$ and $c = 3$ we obtain a maximum temporal range that is two times greater than the TransformerXL with an identical attention cost. Thus if we can learn in the $c > 1$ compressed setting, the temporal range of the model can be significantly increased.

## B  COMPRESSION ACROSS LAYERS

We inspect the compression loss broken down by the layer index, to investigate whether there is a trend in network depth with how compressible the representations are. The compression loss here refers to the attention-reconstruction attention loss. We plot this for a 24 layer trained model on Enwik8, and an 18 layer model trained on WikiText-103. The compression loss for character-based language modelling is about one order of magnitude lower than that of word-level language modelling. The first layer's representations are highly compressible, however from then on there is no fixed trend. Some non-contiguous layers have a very similar compression loss (e.g. 4 & 6, 5 & 7) which suggests information is being routed from these layer pairs via the skip connection.

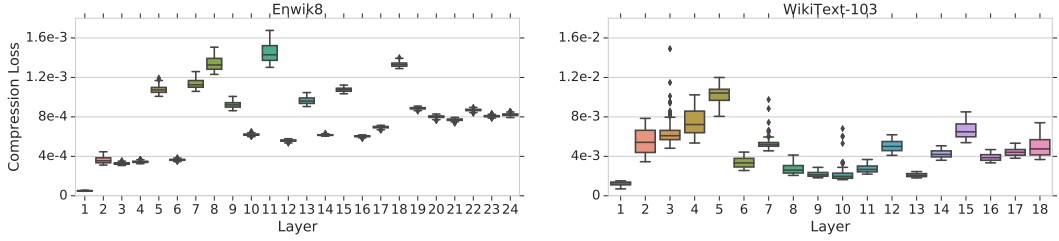

Figure 6: **Model analysis.** Compression loss broken down by layer.

## C  COMPARISON OF COMPRESSED MEMORY SIZES

We compare the best test perplexity obtained for the Compressive Transformer trained on WikiText-103 and Enwik8 across a range of compressed memory sizes. For both models, the best model used a 1D convolution compression network with a compression rate of 3. The Enwik8 model was trained with an embedding size of $1024$, 8 attention heads, 24 layers, an mlp hidden size of $3072$, a sequence window size of $768$, and a memory size of $768$. We see the best compressed memory size is $3,072$ in this sweep, facilitating a total attention window of $3840$. The WikiText-103 model was trained with an embedding size of $1024$, adaptive inputs using the same parameters as (Sukhbaatar et al., 2019), 16 attention heads, 18 layers, an mlp hidden size of $4096$, a sequence window of size $512$ and a memory of size $512$. The best compressed memory size is $1536$ resulting in a total attention window of c. $2048$.

| Compressed Memory Size | 512 | 1024 | 2048 | 3072 | 4096 |
|---|---|---|---|---|---|
| Enwik8 BPC | 1.01 | 0.99 | 0.98 | 0.97 | 1.00 |

Table 8: Compressed memory size vs test performance for Enwik8

| Compressed Memory Size | 256 | 512 | 1024 | 1536 | 2048 |
|---|---|---|---|---|---|
| WikiText-103 Perplexity | 18.2 | 17.9 | 17.6 | 17.1 | 17.7 |

Table 9: Compressed memory size vs test performance for WikiText-103

## D  PG-19 PREPROCESSING

The raw texts from the Gutenberg project were minimally pre-processed by removing boilerplate license text. We then also replaced discriminatory words with a unique ⟨DWx⟩ token using the Ofcom list of discriminatory words [4].

## E  PG-19 TOPICS

We present top-words for some of the topics on the PG-19 corpus. These were generated with LDA topic model (Blei et al., 2003).

Table 10: Examples of top topics on PG-19 corpus.

| Geography | War | Civilisations | Human Condition | Naval | Education | Art |
|---|---|---|---|---|---|---|
| water | people | roman | love | island | work | poet |
| river | emperor | rome | religion | ship | school | music |
| feet | war | greek | religious | sea | life | one |
| miles | army | city | life | men | children | poetry |
| north | death | gods | moral | captain | may | work |
| south | battle | king | human | coast | social | literature |
| mountains | city | first | society | land | child | art |
| sea | soldiers | caesar | man | great | education | great |
| lake | power | great | virtue | found | conditions | poem |
| rock | thousand | romans | nature | islands | well | written |
| mountain | arms | athens | marriage | shore | study | english |
| country | empire | greece | women | voyage | best | author |
| valley | upon | temple | christian | vessels | years | play |
| ice | country | son | age | time | possible | genius |
| west | time | egypt | law | english | class | style |

## F  PG-19 SAMPLES

We show a few different samples from the Compressive Transformer trained on PG-19. We use Nucleus Sampling with $p = 0.98$ (Holtzman et al., 2019). We choose extracts of books from the test set as prefixes. We see the model is able to continue in the style of the text, creating artificial dialogue or descriptive text, and remembering the names of characters over hundreds of words.

As the Compressive Transformer is trained without state resetting, it is actually slightly out of sample when provided with the (relatively) short contexts. This is because its memory and compressed memory may be still empty (whereas they are always full during training). However we see a trend of the samples usually improving towards the end.

---

[4] https://www.ofcom.org.uk/__data/assets/pdf_file/0023/91625/OfcomQRG-AOC.pdf

**Context from Baby Mine by Maragaret Mayo**:

CHAPTER I

Even in college Alfred Hardy was a young man of fixed ideas and high ideals and proud of it.

His friend, Jimmy Jinks, had few ideas and no ideals, and was glad of it, and before half of their first college term had passed, Jimmy had ridded himself of all such worries as making up his own mind or directing his own morals. Alfred did all these things so much better, argued Jimmy, furthermore, Alfred LIKED to do them–Jimmy owed it to his friend to give him that pleasure.

**Compressive Transformer**:

He did not like doing them in the most elegant of all possible marble walks–where freshmen and sophomores stand and talk and laugh and chatter till evening falls and look cool and still and cool and beautifully white in spite of the hot sun, but had to go there for a month before going home. It happened one day that he and Jimmy were strolling together in the green shade of St. Martin's, and Jimmy naturally asked why he was so silent. He was very talkative as most boys are who have once got the habit of it on shore, so the need for answering was soon supplied.

"I am wondering," he said, "why my old friend and chum, Jimmy Jinks, is living a single life and so has no boys. Have you not noticed that many of his old friends had their own boys to follow in their footsteps and help, and often helped successfully too?"

"Oh, yes," said Alfred, "it was plain enough then that he liked to live by himself and got none of his present pleasures or entertainments because of the other fellow's giving it to him."

"Then it IS sad," said Jimmy, "that an old friend like the friend of Big Lights, Jimmy Reynolds, should be living the single life and having no boys."

"And it is pretty true, too," said Alfred, "yet I believe it is the best thing that could happen even now."

"And so do I," said Jimmy, "and I also believe that it is time someone did something to make the things worse for the other's partner."

"And so do I," said Alfred, "and so do I, and so do I."

"What can we do?" asked Jimmy.

"I don't know," said Alfred, "but if someone will come down to the valley and go through a regular mill with the old forces at work, doing the work of the two, and doing it better than any other Miller he ever saw or ever hoped to see, I think there will be a good deal of change in the things and things to be cared for for and for driven and counter-driven."

"I will come and do that mill," said Jimmy.

**Context from The Diary of Samuel Pepys:**

June 1st. Having taken our leaves of Sir W. Batten and my Lady, who are gone this morning to keep their Whitsuntide, Sir W. Pen and I and Mr. Gauden by water to Woolwich, and there went from ship to ship to give order for and take notice of their forwardness to go forth, and then to Deptford and did the like, having dined at Woolwich with Captain Poole at the tavern there. From Deptford we walked to Redriffe, calling at the half-way house, and there come into a room where there was infinite of new cakes placed that are made against Whitsuntide, and there we were very merry. By water home, and there did businesses of the office. Among others got my Lord's imprest of L1000 and Mr. Creed's of L10,000 against this voyage their bills signed. Having wrote letters into the country and read some things I went to bed.

**Compressive Transformer**:

So I rose and did some business at the office, and then to dinner, where great abundance of good things of the kitchen I only being partaker in the feast. After dinner I went to look at the yard and found it in a very bad condition, so I went home and there finding no delight in seeing things I should make shift to get in my way and so I will leave off.

So I took leave of my great friend, and the next day being Sunday, my wife and I and a great number of friends went by water to Deptford, and there took the oaths of allegiance and supremacy and promised such obedience to the Lord as should hereafter turn to the advantage of our State. Being come back we found the shops and warehouses in a pitiable condition, so my wife and I and all the rest of the friends went home and there did the like and by Saturday day I being alone in my office I found myself in a very bad situation, for my Lord, and Mr. Creed, and Pett were all standing ready to receive and receive the oaths of allegiance and right to make use of the ten days' licence I had allowed for this purpose by virtue of the law that was such a help to the servant. Having looked at the ships, I told my wife what I had seen and that I was greatly troubled at my situation and, I hope that will mend it.

So that is the end of a long and patient seeking after those things which in my opinion I deserve and may get if I can only by fair means get them.

**Context from The Patrol of the Sun Dance Trail by Ralph Connor:**

High up on the hillside in the midst of a rugged group of jack pines the Union Jack shook out its folds gallantly in the breeze that swept down the Kicking Horse Pass. That gallant flag marked the headquarters of Superintendent Strong, of the North West Mounted Police, whose special duty it was to preserve law and order along the construction line of the Canadian Pacific Railway Company, now pushed west some scores of miles.

Along the tote-road, which ran parallel to the steel, a man, dark of skin, slight but wiry, came running, his hard panting, his streaming face, his open mouth proclaiming his exhaustion. At a little trail that led to the left he paused, noted its course toward the flaunting flag, turned into it, then struggled up the rocky hillside till he came to the wooden shack, with a deep porch running round it, and surrounded by a rustic fence which enclosed a garden whose neatness illustrated a characteristic of the British soldier. The runner passed in through the gate and up the little gravel walk and began to ascend the steps.

---

**Compressive Transformer**:

As though by mutual consent, all the three members of the Superintendent Force were standing on the veranda watching his progress, a man came strolling slowly down the path, stopping often to gaze curiously about him, then stopping once more when he saw the four standing on the veranda. His face was dark, his eyes prominent, and the lines that furrowed his forehead showed marks of many woes, yet his step was alert and he began to mount the steps with steady strides.

The man's attention was diverted by the unexpected appearance of the two policemen; a third man came slipping up, grumbling under his breath, and stopped long enough to look doubtfully around him, then slipped away again. In the meantime, the four on the veranda continued to watch the approach of the new arrival. In spite of himself, the Superintendent Force became conscious of a curious feeling–uneasiness that was almost fear–in the presence of that strange man. He would not allow himself to admit the fact, yet he felt it in his bones. But to the watchers, the plain, everyday features of that stranger and his coming, seemed only just what the Seven White Shee owed him–their weight, their hurry, their blast.

Had a chariot been made for the good of the three horses, and had the driver been given orders that he should speed them that he might win, they would have been heartening things in the sight of the veteran and the victor. To you they would have been unintelligible to the root of your understanding. When you gaze up in the faces of those four gray horses, you can see clearly through the clouds of dust that rise from their hoofs, and discern plainly where the banker is and where the hobo. Then you will understand why you shall not press the bitter grapes and why you shall not spurn the generous doctrines. You will understand why you shall not praise the lash or the spur, for you will know where the true would be and where the false would be. Then you will understand why you, a man with reason and heart, need not tear your hair over-bitter and why you need not laugh over the blunders of an ignorant man.

About nine o'clock that morning, two buggies, drawn by powerful horses, crossed the Rubicon and turned the railroad from Sandhurst into the Hollow of the Mountains. And though the charioteers stood at their horses' heads, and their drivers cried at their loudest, there was not a man in the four teams who did not feel that his day was worth all the toil and all the peril that he had undergone. And if there were a man in them who did not know that–who did not feel that the road through the Hollow of the Mountains is made easy by the arrival of travelers and by the coming of government, there was one who did not at that moment care whether his day's work were worth all the toil and all the danger that he had had to endure or whether it were not worth more than all.

## AUTHOR CONTRIBUTIONS

Model and Experiment design: JR, TL, AP, SJ
Dataset creation: AP, JR, CH
Text experiments: JR, AP
RL experiments: SJ
Speech experiments: JR

