# OpenReview forum: "Compressive Transformers for Long-Range Sequence Modelling"
_ICLR.cc/2020/Conference — Accept (Poster)_

### Official Review · AnonReviewer3 · 2019-10-22
**Official Blind Review #3**

**Rating:** 8

**Review:**

This paper proposes a way to compress past hidden states for modeling long sequences. Attention is used to query the compressed representation. The authors introduce several methods for compression such as convolution, pooling etc. The outcome is a versatile model that enables long-range sequence modeling, achieving strong results on not only language model tasks but also RL and speech. For testing and evaluating the modeling of really long context sequence modeling, the authors introduce PG-19, a new benchmark based on Project Gutenberg narratives.

The idea is a simple and straightforward one. The choices of compression functions are intuitive and natural. The probably more interesting part of this paper is the training schemes designed to train the memory compression network.

Results are very strong and there is a pretty diverse set of experiments. That said,  it seems like a huge amount of resources were spent on this work alone. It also seems like these models are not trivial to train (or get them to work). It would be interesting to find out how much resources were spent (in terms of preliminary experiments) to getting these models to start working decently. There are also no reports of parameter counts, which might make the experiments unfair.

Achieving SOTA is one thing, which could be attributed to large resource pools and maybe larger parameter sizes of models.

Overall, I am voting for a weak accept. While this paper is more incremental and novelty may be slightly lacking, I think the breadth of experiments and competitive results warrants an acceptance.

Several issues and questions for the authors:

1) Why are the results on PG-19 not reported in a Table format? Why are there no results of the base Transformer on PG-19? I think this is really necessary and should be reported.
2) The authors mention that this memory compression architecture enables long sequence modeling. However, is there an intended way of use for long-text that is not necessarily framed as a LM problem? For instance, results on NarrativeQA benchmark would be nice.

UPDATE: I have read the author response and other reviewer's comments. I am happy with the efforts made by the authors and I am raising my score to 8 (accept).



**Experience Assessment:**

I have published in this field for several years.

**Review Assessment: Checking Correctness Of Derivations And Theory:**

N/A

**Review Assessment: Checking Correctness Of Experiments:**

I assessed the sensibility of the experiments.

**Review Assessment: Thoroughness In Paper Reading:**

I read the paper at least twice and used my best judgement in assessing the paper.

---

> ### Author Response · Authors · 2019-11-10
> **Re resources, training difficulty, other text applications, PG-19 results**
>
> Thank you for your thorough review!
>
> Re. “It would be interesting to find out how much resources were spent (in terms of preliminary experiments) to getting these models to start working decently.”
>
> The majority of experiments were spent reproducing the sota (at the time) TransformerXL; that is, getting the model and training setup working well. We plan to open-source the TransformerXL baseline alongside the Compressive Transformer in TensorFlow (The TXL is now open-sourced in a few locations also). We considered 7 model/loss compressive transformer variants, displayed in Table 4, and ran 16 experiments in total on enwik8. These experiments swept over compression rates (typically 1-4) and then we experimented with different model setups. We then ran 6 compressive transformer experiments on WikiText-103.
>
> Re “It also seems like these models are not trivial to train (or get them to work)”
> We trained these models with the same parameters as the transformerxl and we basically found (as shown in Table 4) that pretty much all compression approaches worked ok. Even mean-pooling activations performed reasonably (exceeded baseline performance and matched the current sota). However the learnable conv1d performed the best. The optimization schedule of decreasing optimization updates (S5.6.3) allowed us to achieve better results but this wasn’t necessary to train the models. So we would challenge the conclusion that this model is difficult to train.
>
> Re.  is there an intended way of use for long-text that is not necessarily framed as a LM problem? … Such as NarrativeQA
>
> We think any sequential prediction problem with long-range dependencies is a good fit for this model. Ideally a streaming task where you need to maintain an online representation of the past that is quickly updated. So perhaps reading comprehension tasks where you read a book but periodically answer questions about it, a little like Children’s Book Test but with longer contexts. For summarization, such as NarrativeQA, only one set of predictions needs to be made at the end of the book and it appears that the best solutions are (currently) maintaining the book statically in a simple embedded space and repeatedly attending to it, possibly copying sections of text. It would be interesting to see the results from simple autoregressive models for summarization nonetheless.
>
> Re. Why are the results on PG-19 not reported in a Table format?
>
> Very good point. We have remedied this, it is now in a table. We also have new results with larger models that serve as a better initial baselines
>
> 36 layer TransformerXL (3,000 mem) 					                36.25
> 36 layer Compressive Transformer (1,500mem + 1,500 CM)		33.6

---

### Official Review · AnonReviewer2 · 2019-10-22
**Official Blind Review #2**

**Rating:** 8

**Review:**

## Updated review

I have read the rebuttal. First I'd like to thank the authors for the detailled rebuttal.
The latest version of the paper adressed all my concerns, hence I change my rating to Accept.

## Original review

This paper presents a new variation of the Transformer model, named Compressive Transformer. The key novelty of this model is to preserve long range memory in a compressed form, instead of discarding them as previous models have done. This improves the long-range dependencies modelling capabilities of the approach. The model is evaluated on two common language modelling benchmarks and yields state of the art results in both of them. The paper also introduces a new benchmark for long-range dependencies modelling composed of thousands of books. The paper finally presents an analysis of the compressed memory and provide some insights, including the fact that the attention model uses the compressed memory. The model is also evaluated on two other tasks: speech generation and reinforcement learning on videos.

I think this paper should be accepted, mainly because:
- The proposed model is novel as far as I can tell.
- The presented approach is significant, as modelling long-range dependencies is an important milestone in sequence modelling.
- The new benchmark is a good addition.
- The comparison with the relevant literature is thorough and well done.
- The experiments are convincing and demonstrate the viability of the approach, although some aspects can be improved (see below).

Detailed comments:
- About the character-level language modelling on Enwik8, the improvement is very small, it seems that the task doesn't benefit from have long-range memory, could it be because character-level modelling is less dependent on the long-range past? can the authors comment on that? It would also been interesting to evaluate the gain of the memory, for instance by varying the size of the compressed memory from 0 to 1152.
- The WikiText-103 evaluation is interesting, specially Table 6, which shows the advantages of the model. However when comparing with the literature, it's not clear if the performance gain is due to the compressed memory or to the network capacity. A study with different lengths of the compressed memory (starting at 0) would bring some insights about that.
- In Section 5.6.2: can the authors justify why the attention weights were split in only 6 bins? creating a trended curve on only 6 points could be problematic, and I don't see why more bins couldn't be used.
- The speech analysis section (5.7) is not very insightful. It shows that the proposed model is on par with WaveNet on unconstrained speech generation, which is not very useful and feels a bit half-finished. I think that the authors should either commit to this study by constraining the model with linguistic features like in (Oord et al. 2018) and evaluate it in a TTS framework with subjective evaluation or discard this section entirely.


**Experience Assessment:**

I have read many papers in this area.

**Review Assessment: Checking Correctness Of Derivations And Theory:**

N/A

**Review Assessment: Checking Correctness Of Experiments:**

I assessed the sensibility of the experiments.

**Review Assessment: Thoroughness In Paper Reading:**

I read the paper at least twice and used my best judgement in assessing the paper.

---

> ### Author Response · Authors · 2019-11-10
> **Re model ablations, enwik8, attention weights & speech modelling**
>
> Thank you for your kind review!
>
> Regarding memory size: here’s an ablation with performance versus compressed memory size for both enwik8 and wikitext-103! Both models improve significantly as a function of compressed memory size from small values. There is an optimal value, if we make the compressed memory much larger than the training regime then performance eventually deteriorates as the model’s attention drifts out-of-distribution (e.g. 4096+ for Enwik8). We have added this table to the paper also.
>
> Enwik8
> Compressed Memory Size	         512	       1024	         2048	3072	4096
> BPC				                         1.01	0.99	         0.98	0.97	        1.00
> (Model has a chunk size of 768 and memory size of 768)
>
> WikiText-103
> Compressed Memory Size	  256	512	         1024	1536	2048
> Perplexity			         18.2	17.9	         17.6	17.1	        17.7
> (Model has a chunk size of 256 and memory of size 512)
>
> Note that CM=0 is literally the TransformerXL which we have included results for in the paper (incl. our implementation). For the published TransformerXL’s 18.3 perplexity, it was using an attention window of 1600 but we improve on this result with an attention window of only 768 (512 + 256).
>
> Re Enwik8: We agree the improvement on Enwik8 may seem quite small but this is partially due to the metric. BPC has a very small range. If we look at the word-level perplexity of these models, the 0.99bpc transformerxl has a word-level perplexity of 170 whereas the 0.97bpc sota compressive transformer has a word-level perplexity of 153. So a gain of 17 perplexity. This calculation comes from ppl_word = 2^(7.48 * bpc) as 7.48 is the average word-length in enwik8’s test set. Enwik8 actually has a longer range of dependency over wikitext-103 because of the more granular sequence data; they both represent wikipedia pages but processing the article at the character-level stresses the model’s range of attention.
>
> Re speech: It would be preferable to perform a full human quality survey. The observation we wanted to convey was that one can get a transformer-like model to model high-frequency speech unconditionally and the compressive model helped in obtaining learning dynamics that are comparable with wavenet (in comparison to the TransformerXL which performs worse).
>
> However we do not wish to claim that this implies we have a better text-to-speech model; this would require substantially more work, conditioning on linguistic features, and expert human raters. Instead of focusing on text-to-speech, we look at raw speech modelling which has many downstream applications beyond text-to-speech (e.g. speaker identification) and stresses long-range dependency. We have made this more clear in the text (update soon-to-be-posted), and will consider removing the results entirely if other reviewers feel the experiment is misleading.
>
> Re. why six attention bins? We just chose a multiple of 3 (so the buckets have boundaries at the compressed_memory, memory, sequence boundaries) that is not too large such that there’s not too much noise. However we have re-run this analysis with 18 buckets and are including the updated figure in our (soo to be posted) updated paper. This is a better visualization of the data and captures the trend more carefully (we also remove the trend curve and switch to violin plots to better display the variability of each bucket). However the conclusion remains the same - that there is an increase in attention weight over the compressed memories versus the older regular memories

---

### Official Review · AnonReviewer1 · 2019-10-23
**Official Blind Review #1**

**Rating:** 6

**Review:**

This paper investigates a so-called "compressive transformer" approach. The idea is to compress distant past memories into a coarse-grained representation while keeping a fine-grained representation for close past memories.  A variety of compression techniques and training strategies have been investigated in the paper and verified using tasks from multiple domains including language modeling, speech synthesis and reinforcement learning. Particularly, the authors propose a new benchmark PG-19 for long-term sequence modeling.

Overall, I found the work interesting and experiments are thorough and strong.   It is always great to see a new benchmark released to the community.  That being said, I have concerns regarding the paper.  The authors put huge amount of effort into the experiments but only describe the proposed technique in a very rough and abstract way, lacking necessary technical details to formulate the technique. What is the mathematical formulation of the problem?  How exactly the compression is carried out on various network architectures is not clear after reading the paper.  Also, I guess many readers including me do not have a perfect understanding of Fig. 1 although it shows something intuitively. (What is the difference between different colors? What is the difference between sequence, memory, and compressed memory?  What do the arrows mean? There is no explanation whatsoever either in the figure or in the caption).  This is the major concern I have regarding the paper.  Despite of the strong experimental presentation, lacking the technical details has significantly hurt the quality of the paper.

P.S.  Thanks for the rebuttal.  I have lifted my score.

**Experience Assessment:**

I have read many papers in this area.

**Review Assessment: Checking Correctness Of Derivations And Theory:**

N/A

**Review Assessment: Checking Correctness Of Experiments:**

I assessed the sensibility of the experiments.

**Review Assessment: Thoroughness In Paper Reading:**

I read the paper at least twice and used my best judgement in assessing the paper.

---

> ### Author Response · Authors · 2019-11-10
> **Re technical detal**
>
> We completely agree that the model could be described more explicitly. *We are updating the paper with more mathematical details and an algorithm box to make things more explicit*. We originally wrote this paper to convey the key components of the model for those familiar with TransformerXLs, with the idea that all of the fine details are better represented in the code --- however we realize this was not the best strategy. We will still open-source the code so people can use the model and be certain of every detail, but we are completely re-writing the model section with the inclusion of an algorithm box. As pseudo-code here, the compression mechanism is really just passing memories that would otherwise be forgotten through a conv1d compression network:
>
> compression_rate <- 3
> old_memory  <- memory[:-seq_size]  # the memories to be forgotten
> compression_fn <- conv_1d(kernel_size=compression_rate, stride=compression_rate)
> new_cm <- compression_fn(old_memory )  # new compressed memories
>
> Then for attention, before in the TransformerXL one would compute
> attention(seq, [memory, seq])
> whereas here we compute
> attention(seq, [compressed_memory, memory, seq])
>
> Before in the TransformerXL one would update memory by concatenating the sequence and truncating the oldest memories (to keep the memory fixed-size):
> memory <- concat_and_truncate(memory, sequence)
>
> where 'concat_and_truncate' refers to:
> def concat_and_truncate(old_state, new_state):
>     new_state_size <- new_state.shape[1]  # time dimension
>     return concat([old_state, new_state])[new_state_size:]
>
> Now we update both the memory and compressed_memory:
> memory <- concat_and_truncate(memory, sequence)
> compressed_memory <- concat_and_truncate(compressed_memory, new_cm)
>
> In Figure 1 we kept the sequence and memory the same colour, as these hidden activations represent information for a single time-step in the transformer. We use an arrow to indicate that we map a set of memories to a smaller set of compressed memories. We chose a different colour for the compressed memories (and made the ticks more frequent) to indicate that these represent information over multiple time-steps. We are updating the figure and caption with more details such that this is clearer.
>
> If there is anything else that is unclear, feel free to give us feedback!

---

### Public Comment · ~Aran_Komatsuzaki1 · 2019-10-02
**(Self-)attention for compression**

I believe you could've instead used (self)-attention to produce compressed memory. Or is this not viable for some reason?

---

> ### Public Comment · ~Artus_KG1 · 2019-10-02
> **Re:**
>
> Seems possible but would be more costly (square vs linear). Would indeed be interesting to see if attention on data compressed with attention is better than attention on data compressed by convolutions is more effective.
>
> 100 TPUv3 cores is a nice bunch of compute.

---

> > ### Author Response · Authors · 2019-10-22
> > **attention for compression**
> >
> > Thanks so much for your comments and innovative line of thinking. This is something we have considered but had not come around to trying!
> >
> > I don't think it's infeasible, however there are several ways of using attention for compression, some of them are not desirable.
> >
> > E.g. if one has n memories to compress to n/c compressed memories. One could instantiate n/c learnable parameters, each performs attention over the memories to compress. This would certainly result in n/c compressed memories, where attention was used to perform the compression. This scheme could be effective, but it makes the scheme dependent on the memory size.
> >
> > Another idea was to use the conv1D, or even pooling, to reduce the number of memories (from n -> n/c) and then do self-attention over this set to absorb information across. We did not try this but it seems reasonable. Conversely we could perform self-attention over the memories and then compress, we think this would be powerful but too expensive as it would effectively double the compute of the whole model.
> >
> > If there's anything obvious we are missing, feel free to comment!

---

> > > ### Public Comment · ~Aran_Komatsuzaki1 · 2019-10-24
> > > **Some thoughts**
> > >
> > > Thank you very much for your response. It may be hard to compensate for the computation overhead due to self-attention at this moment, but I'm very excited for further development in this topic. I was also trying to make the context of Transformer unlimited as in this study (nevertheless not succeeded).
> > >
> > > One thing I think is worth considering is, by generalize the idea of [1], to let the model to occasionally predict a randomly sampled segment of (either near or distant) past sequence ("distant" seq. comes from outside of the current TBPTT segment). Successful memory architecture should be able to recall the past easily, so this may become an alternative to measure how far your architecture can track back. Also, this may improve the retention of memory.
> > >
> > > [1] Learning Longer-term Dependencies in RNNs with Auxiliary Losses https://arxiv.org/abs/1803.00144

---

> > > > ### Author Response · Authors · 2019-10-24
> > > > **re. Predicting the past**
> > > >
> > > > This is a good point, one room for improvement is further analysis of whether the model's temporal range is indeed increased. The greater relative improvement in prediction of rare words (vs frequent words) hints that the performance improvement is due to longer-range reasoning, but it would be nice to make this more explicit. We could fine-tune a trained model on the task of predicting the past at varying intervals to see how it compares to the TXL. If we get time to perform this analysis before the discussion period is over, we will include these results.

---

### Public Comment · ~Sainbayar_Sukhbaatar1 · 2019-10-04
**model sizes**

What are the model sizes (the total number of parameters, hidden sizes, the number of heads, etc.) used in the language modeling experiments? I can't find it in the paper.

---

> ### Author Response · Authors · 2019-10-22
> **re. model sizes**
>
> We wanted to use the exact model setup from the TransformerXL, which we used as our baseline. So for WikiText-103 this was 18 layers with a hidden size of 1024 (16 heads), 4096 mlp hidden size, using the same adaptive input representations scheme to embed words. For Enwik8 we used the same 24 layer model, 1024 embedding and hidden size, 8 heads, 3072 mlp hidden size.
>
> In terms of the number of parameters optimizing the loss, this is exactly the same as the TransformerXL 277M for Enwik8 and 257M for WikiText-103.
>
> For the compression network, which was only optimized with respect to the auxiliary compression loss, this consumed 0 params for max/mean pooling, and most-used. For 1D conv it consumed 1M x compression_rate x #layers params, and for the dilated convolution it consumed more. We will update the paper with much more explicit model details since this is clearly a room for improvement.

---

> > ### Public Comment · ~James_A_Bowery1 · 2020-01-26
> > **Commensurability of Table 4 Items**
> >
> > The judging* criterion for the Hutter Prize is size of a self-extracting archive of the enwik8 corpus, to standardized on the algorithmic resources available to the archive.  This is essential for commensurability under the principle of minimum description length (MDL) approximation of Kolmogorov Complexity.  Dividing the corpus into training and testing sets is neither necessary nor desirable under this metric.
> >
> > Controlling for the same "model setup" is a big step in the right direction -- as it increases the commensurability with TransformerXL -- particularly as compared to the other items in Table 4.  While model ablation can produce even more commensurable measures, it would be helpful for SOTA comparisons to be more rigorous in defining the algorithmic resources assumed in their measurements.
> >
> > A consequence of improved rigor would be to expose just how important "small" improvements, such as .99 to .97 can  be, as indeed they are.
> >
> > *I'm on the Hutter Prize judging committee.

---

> > > ### Author Response · Authors · 2020-01-27
> > > **RE**
> > >
> > > I think I agree with a lot of your comment. Just to be clear, although we use enwik8 as a dataset for language modelling, we have no stake in the Hutter Prize. This model, along with pretty much all neural network language models trained on this dataset, are too large to be competitive with the algorithms devised by Rhatushnyak. If the prize had been devised using 10GB of wikipedia then it would be a different story. There are lots of tricks to cut the final parameter count (e.g. make some of the linears low-rank, prune the weights, distill the large model to a smaller model etc.) if one wants to benchmark models at a fixed parameter budget. Our opinion is that it's a worthwhile pursuit to see what language model generalizes best irrespective of parameter size. Simply scaling the transformerxl to a larger no. parameters via larger width or a larger number of layers did not improve generalization.

---

> > > > ### Public Comment · ~James_A_Bowery1 · 2020-01-27
> > > > **Principled As Well As Practical SOTA Benchmarking**
> > > >
> > > > I agree that the time is long-since past for a prize based on an expanded corpus, such as the one you are (soon?) going to publish from Project Gutenberg. This should have been done by someone with deep-pockets, i.e. Google with the advent of the one billion word benchmark, because it came years after Hutter's enwik8 prize.
> > > >
> > > > But I would strongly suggest that any move forward toward such a benchmark specify the minimum _practical_ common algorithmic resources (Universal Turing Machine instruction set) upon which to run a decompression program that produces the benchmark corpus.
> > > >
> > > > By "_practical_" I admit that nowadays it may only be _practical_ to assume an "instruction set" consisting of the entire Tensorflow API -- particularly for a DeepMind-financed benchmark prize.
> > > >
> > > > As for seeing what language model generalizes best, there are two quite distinct levels to this question:
> > > >
> > > > 1) Empirical testing of the MDL principle in SOTA claims.
> > > > 2) Application of the MDL principle in SOTA claims.
> > > >
> > > > Philosophically, the MDL principle is already assumed in virtually all science and engineering due to "the unreasonable effectiveness of mathematics in the natural sciences."  So, on that basis, #1 is similarly assumed by those, such as Hutter, who finance #2, as would a DeepMind prize based on size of self-extracting archive.
> > > >
> > > > #1  is for who don't place their "faith" in such philosophical arguments, and is where tests, such as yours based on division of test and training sets, can do more than just measure "what generalizes best":  They can, through model compression/model ablation/knowledge distillation, see if MDL* holds as a meta-empirical truth.
> > > >
> > > > See "Extreme Language Model Compression with Optimal Subwords and Shared Projections"
> > > >
> > > > https://arxiv.org/abs/1909.11687
> > > >
> > > > *By "MDL" I am here assuming UTM algorithmic capacity in the description's language.

---

### Author Response · Authors · 2019-11-12
**Updated paper**

Thanks for the comprehensive reviews, they have certainly improved the quality of the paper.

List of changes:

[credit to reviewer 1]
- Updated figure 1 and caption with more details.
- Re-written model section: added formal notation, added algorithm box for full model, and for attention-reconstruction loss.
- Added subsection on temporal receptive field.

[credit to reviewer 2]
- Attention bins are more granular, include uncertainty over attention per bucket. Remember, the self-attention is causally masked (mentioned in the text) thus the increase in attention to earlier sequence. Crucially, there is an increase in attention from the oldest memories, to the newest compressed memories (which are older).
- Added memory size ablations (Table 8 & 9).

[credit to reviewer 3]
- Added PG-19 results table with Compressive Transformer and TransformerXL (improved both models from original result, using deeper networks).

We appreciate the reviewers have a limited time to read paper revisions, however we feel almost all points have been substantially addressed and thus we would strongly welcome feedback.

---

> ### Author Response · Authors · 2019-11-13
> **^**
>
> Fixed some typos and further clarified algorithm box in paper update. Please feel free to scan over the revised text and express any other points of concern!

---

### Decision · Program_Chairs · 2019-12-19

**Decision:**

Accept (Poster)

**Comment:**

The paper proposes a "compressive transformer", an extension of the transformer, that keeps a compressed long term memory in addition to the fixed sized memory.  Both memories can be queried using attention weights.  Unlike TransfomerXL that discards the oldest memories, the authors propose to "compress" those memories.  The main contribution of this work is that that it introduces a model that can handle extremely long sequences. The authors also introduces a new language modeling dataset based on text from Project Gutenberg that has much longer sequences of words than existing datasets.  They provide comprehensive experiments comparing against different compression strategies and compares against previous methods, showing that this method is able to result in lower word-level perplexity. In addition, the authors also present evaluations on speech, and image sequences for RL.

Initially the paper received weak positive responses from the reviewers. The reviewers pointed out some clarity issues with details of the method and figures and some questions about design decisions. After rebuttal, all of the reviewers expressed that they were very satisfied with the authors responses and increased their scores (for a final of 2 accepts and 1 weak accept).

The authors have provided a thorough and well-written paper, with comprehensive and convincing experiments. In addition, the ability to model long-range sequences and dependencies is an important problem and the AC agrees that this paper makes a solid contribution in tackling that problem.  Thus, acceptance is recommended.